# Responses to Reduced Feeding Frequency in Captive-Born Cheetahs (*Acinonyx jubatus*): Implications for Behavioural and Physiological Stress and Gastrointestinal Health

**DOI:** 10.3390/ani13172783

**Published:** 2023-08-31

**Authors:** Kelsey Lee Brown, André Ganswindt, Gerhard Steenkamp, Adrian Stephen Wolferstan Tordiffe

**Affiliations:** 1Department of Paraclinical Sciences, Faculty of Veterinary Science, University of Pretoria, Pretoria 0110, South Africa; 2Centre for Veterinary Wildlife Studies, Faculty of Veterinary Science, University of Pretoria, Pretoria 0110, South Africa; 3Department of Zoology and Entomology, Faculty of Natural and Agricultural Sciences, Mammal Research Institute, University of Pretoria, Pretoria 0028, South Africa; 4Department of Companion Animal Clinical Studies, Faculty of Veterinary Science, University of Pretoria, Pretoria 0110, South Africa

**Keywords:** cheetah, captive diet, wildlife husbandry, gastrointestinal health, stress

## Abstract

**Simple Summary:**

This study examined how offering larger quantities of food less frequently to better replicate their natural feeding pattern could affect the health of captive-born cheetahs. For three weeks, six hand-reared cheetahs were fed four once-daily meals per week, followed by three weeks in which they were fed two daily rations six days a week for the same duration while maintaining their total weekly food intake. The studied cheetahs showed higher faecal consistency scores and activity levels when fed less frequently. The results indicate that reducing feeding frequency could benefit captive cheetahs’ gastrointestinal health without causing significant stress.

**Abstract:**

Unnatural diet composition and frequent feeding regimes may play an aetiological role in the multiple diseases prevalent in captive cheetahs. This study investigated the responses of captive-born (hand-reared) cheetahs (*n* = 6) to a reduced feeding frequency schedule distinguished by offering larger quantities of food less frequently. The study cheetahs were fed four once-daily meals per week during the 3-week treatment period, followed by a 3-week control period in which they were fed two daily rations six days a week. Total weekly food intake was maintained throughout the study. Variations in behaviour, faecal consistency score (FCS), and faecal glucocorticoid metabolite concentration were measured. Less frequent feeding resulted in higher FCS (*p* < 0.01) and locomotory behaviour (*p* < 0.05) among the studied cheetahs. Faecal glucocorticoid metabolite concentration demonstrated an initial acute stress response to the change in feeding frequency (*p* < 0.05) and subsequent adaptation. The results of the FCS analysis suggest that the more natural feeding pattern could have benefited the studied cheetahs’ gastrointestinal health without a significant behavioural or physiological stress response overall to the change in feeding frequency.

## 1. Introduction

In the wild, cheetahs (*Acinonyx jubatus*) typically prey on small to medium-sized antelope species with a body mass of between 23 and 56 kg [1] and, if left undisturbed, can consume a large proportion of the carcass [2]. They seldom eat daily or at a fixed interval; cheetahs exhibit a feeding pattern alternating between consuming large meals and periods of limited or no food influenced by irregular prey availability in their natural habitat [3]. However, captive cheetahs are routinely fed a nonvarying diet of skinned muscle meat from livestock species, commercially prepared carnivore diets, carcass parts, or a combination [4,5] offered at fixed intervals once or twice daily, with only one fast day per week. A whole carcass diet of the cheetah’s natural prey species and the associated feeding habits are challenging to replicate in captivity [6]. Some facilities argue that frequent feeding allows for daily monitoring of the animals’ appetites as an indicator of health and reduces boredom-induced stress [7]. However, unnatural diet composition and frequent feeding regimes may play an aetiological role in the prevalence of gastrointestinal (GI) and metabolic diseases in captive cheetahs [5,8].

One of these diseases, *Helicobacter*-associated gastritis, causes significant morbidity and mortality in captive cheetahs worldwide [9,10,11]. *Helicobacter* species, spiral bacteria colonising the stomach, infect most captive and wild cheetahs [12]. Captive cheetahs typically have some degree of inflammation (gastritis) that can be asymptomatic or associated with regurgitation, vomiting, the passage of undigested food, and weight loss [13,14]. However, in free-ranging cheetahs, there is colonisation by abundant spiral bacteria but little to no associated inflammation [10,15], demonstrating the likely multifactorial aetiopathogenesis of gastritis.

Diet, as a potential risk factor for GI pathology in captive cheetahs, was previously dismissed for the most part [10]. More recently, Whitehouse-Tedd et al. [5] found that feeding horsemeat has a significant (detrimental) relationship with gastritis risk in captive cheetahs. They attributed this to its high protein content [5] and/or digestibility [16] relative to other meat types fed to captive cheetahs. Crude protein is anticipated to be high in all carnivorous diets, including that of free-ranging cheetahs. However, the frequency with which it is fed in captivity and its quality could affect GI health by changing the amount of protein reaching the large intestine. Colonic fermentation of poorly digested dietary protein modifies microbiota composition in favour of proteolytic bacteria, some of which can be pathogenic in high concentrations [17,18] and produce putrefactive compounds (e.g., ammonia, indoles, phenols) associated with various disease states [19,20,21]. Moreover, horse (in particular) is commonly fed as muscle meat without low to nondigestible collagen-rich matter (e.g., bone, tendons, cartilage); therefore, its relative lack of ‘animal fibre’ may further increase putrefaction of digesta in the intestine [19,20,22,23].

Transforming gut bacteria-derived putrefactants into toxic metabolites negatively affects multiple other organ systems and metabolic pathways. Uraemic toxicity of indoxyl sulphate is associated with the progression of chronic renal failure [24], a significant cause of death in captive cheetahs [25,26,27,28]. The renal lesions in captive cheetahs resemble diabetic glomerulopathy in humans and chronic progressive nephropathy in rats [25]. High-protein diets, particularly when fed ad libitum and continually, accelerate glomerulosclerosis in rats [29,30,31] and could be a comparable dietary risk factor for kidney damage in captive cheetahs.

A growing body of evidence suggests that feeding restrictions shape the gut ecosystem, function, and interaction with the host. Intermittent fasting has beneficial regulatory effects on immune homeostasis and intestinal microbiota composition in human and rodent models [32,33,34,35]. Furthermore, intermittent fasting attenuates the colon tissue inflammatory response and oxidative stress [32,36]. Following the adaption of captive lions (*Panthera leo*) from a conventional zoo feeding programme of predictable, fixed, small daily meals to a more natural gorge and fast feeding schedule of larger, more infrequent meals, Altman et al. [37] reported improved digestibility of a horsemeat-based diet. Considering similarities in the species’ natural feeding ecology, fasting conditions could have digestive health benefits for the cheetah as they did for the lion.

This study’s overall aim was to investigate the responses of captive-born (hand-reared) cheetahs to a reduced feeding frequency schedule distinguished by offering larger quantities of food less frequently. We hypothesised that a more natural feeding pattern would beneficially impact the GI ecosystem, including the microbial fermentation process. In previous studies, faecal consistency scoring has been used as a noninvasive method of measuring GI health in cheetahs, other exotic felids [5,38], and domestic carnivore species [39,40]. We also hypothesised that changing the feeding frequency may result in a behavioural and/or physiological stress response. Poor faecal consistency has been linked to GI stress in captive carnivores [38]. Behavioural observations [41] and faecal glucocorticoid metabolite (fGCM) analysis [42] were used as more established stress-related markers in the cheetah. In addition, we explored the use of biologging technology to record body temperature (T_b_), heart rate (HR), and locomotor activity (LA) simultaneously (refer to Appendix B). We predicted that higher fGCM concentrations, T_b_, and HR would indicate a physiological stress response.

## 2. Materials and Methods

### 2.1. Study Site and Animals

The experimental trials took place between April and September 2019 at the Cango Wildlife Ranch and Conservation Centre (33°33′ S, 22°12′ E), 4 km north of Oudtshoorn, a semiarid region in the Western Cape of South Africa. Study months (autumn and mainly winter) were distinguished by short photoperiods and cold air temperatures (T_a_), ranging from 13 to 17 °C.

Three male (CH-2205, -2206, and -2271) and three female (CH-2207, -2276, and -2277) adult cheetahs (Table 1) habituated to human presence and interacting daily with the facility’s caretakers in the absence of restraint were assigned to this study. The study cheetahs were housed off-exhibit at the Jill Bryden-Fayers Reserve, neighbouring the Cango Wildlife Ranch. They were held in outdoor enclosures ranging from 400 to 1350 m^2^, adjoining conspecifics. The enclosures’ topography was varied and naturalistic, consisting of a dirt substrate, vantage points and marking areas (e.g., rocks, tree stumps), sufficient vegetation to hide, and a wooden shed for shelter. Enclosures were cleaned once or twice daily.

### 2.2. Experimental Design

The experimental trials commenced following a ≥12-day washout after surgical implantation of the biologgers (refer to Section A.1.2. Surgical procedures). In a pilot study conducted on the study cheetahs, T_b_, HR, LA, faecal consistency score (FCS), and fGCM concentration data demonstrated temporal recovery by postoperative day 10 [43]; therefore, the authors felt ≥12 days after surgery to be sufficiently long to commence the experimental trials. They were conducted using a within-subject experimental design, where each cheetah in the study served as their own control. Grouped by their respective ages, the study cheetahs received the treatment, i.e., a reduced feeding frequency schedule in an initial 3-week period followed by a 3-week control period, against which the effects of the treatment were measured (Table 2).

At the Cango Wildlife Ranch, cheetahs are fed a horsemeat-based diet prepared on-site, weighed, and recorded. To supplement the nutritional composition of horsemeat, i.e., moisture: 71.9%, dry matter: 29.1%, crude protein: 19.8%, and crude fat: 6.63% [44], 7.5 g of predator powder (V-Tech Pty Ltd., Midrand, South Africa; containing 35 g of calcium per 100 g of powder) and 20 g of glycine (WildCat Nutrition Pty Ltd., Pretoria, South Africa) is added per 1 kg of meat [45]. Regarding paired housing (Table 1), 1 tbsp of uncooked (nondigestible) rice was thoroughly mixed into the diet of study cheetahs CH-2206 and CH-2277 once daily to assign individual faecal samples—the facility’s caretakers separate cheetahs during feeding to reduce competition for food and prevent meal sharing. Meals are offered at variable intervals to prevent food anticipatory behavioural activity [46]. Once weekly at random, cheetahs are fed horse shank or rib bones with some meat intact equivalent to the weight of their daily ration in place of the day’s meals to maintain variety and provide periodontal stimulation. The bones are not consumed by the cheetahs and are removed and discarded. Leftover meat is removed, weighed, recorded, and discarded. Water is available ad libitum.

During the 3-week treatment period, the study cheetahs were fed on a reduced feeding frequency schedule, where meals were offered once daily between 08.00 and 17.00 h, Monday, Tuesday, Thursday, and Friday (Table 2). Weekly fasting days were assigned to Wednesday, Saturday, and Sunday. Larger than regular meals were offered on feed days to maintain total weekly food intake despite additional fast days. The three-year-old study cheetahs (CH-2205, -2206, and -2207) were fed 2.7 kg per day four days a week, and the two-year-old study cheetahs (CH-2271, -2276, and -27) were fed 2.5 kg per day four days a week. During the following 3-week control period, the study cheetahs were fed on a feeding schedule routinely used at the Cango Wildlife Ranch, where meals were offered twice daily between 08.00–12.00 and 15.00–17.00 h, Monday to Saturday. Weekly fasting days are assigned to Sundays. The three-year-old study cheetahs were fed 1.8 kg per day portioned into two rations six days a week, and the two-year-old study cheetahs were fed 1.6 kg per day portioned into two daily rations six days a week. There was a washout between the treatment and control, during which the study cheetahs were fed on the routine feeding schedule.

Other than the specific intervention being investigated, i.e., a reduced feeding frequency schedule, the study cheetahs’ environment, housing, and management (refer to Section 2.1. Study Site and Animals) were maintained across the treatment and control, including bones offered randomly once weekly in place of the day’s meals.

Throughout the study, i.e., in the treatment and control, each cheetah was monitored regarding (i) behaviour, (ii) FCS, and (iii) fGCM concentration.

### 2.3. Behavioural Data Collection

Each cheetah in the study was observed 15 times during the treatment and control, respectively. These observations were conducted between 07.00 and 17.00 h, Monday to Sunday, within the operating hours of the Cango Wildlife Ranch. During five weekly 60 min observation sessions, the principal investigator (KLB) carried out 12 instantaneous scan samples [47] with a 5 min interscan interval per enclosure. Regarding paired housing (Table 1), the study cheetahs were observed together using physical identifiers to assign individual behaviour. Sampling was conducted on a variable day-and-time basis between enclosures, randomly selected to prevent time-of-day effects.

The study recorded 15 behaviours categorised as ‘inactive’, ‘active’, and ‘not observed’ (Table 3), informed by the previous literature on felid behaviour (specifically cheetahs) [37,48,49,50] and initial observations of the cheetahs being studied. Time spent out of sight (hiding or staying away from the human observer) was noted as its performance has been linked to a psychological stress response in felids [51,52,53,54,55].

### 2.4. Faecal Sample Collection and Consistency Scoring

During the operating hours of the Cango Wildlife Ranch (08.00–17.00 h), faeces were collected within one hour after defecation. Faeces excreted between 17.00–08.00 h, when the study cheetahs’ enclosures could not be entered, were collected within 16 h after defecation (4–10 °C T_a_) [56].

Following sample collection, the principal investigator (KLB) assigned FCS using a five-point faecal scoring system adapted from that developed by Whitehouse-Tedd et al. [5]. In this study, the five-point faecal scoring system used (grades ranging from 1 to 5, where grade 1 was the lowest and grades 2–5 were progressively higher) included two points (grade 4: firm and dry, and grade 5: firm) considered to be ‘normal,’ and three points (grades 1–3: liquid, soft without shape, and soft with shape) considered to be ‘suboptimal’ according to free-ranging cheetah scat. As a species inhabiting semiarid regions (such as Oudtshoorn), dry faecal consistency was not considered dissimilar to faeces found in free-ranging cheetahs [45].

Afterwards, the samples were deposited into appropriately labelled (sample collection date, study cheetah, and sample identification number) 50 mL polypropylene specimen containers and frozen at −20 °C.

### 2.5. Faecal Steroid Extraction and Quantification

Following completion of the experimental trials, faecal samples were transported frozen to the Endocrine Research Laboratory, University of Pretoria, South Africa. Faecal steroids were extracted and subsequently analysed for fGCM concentration.

Frozen faecal samples were lyophilised, and the resultant dry faeces were pulverised and sieved through a mesh strainer to remove fibrous material [57]. Between 0.050 and 0.055 g of faecal powder was weighed per sample and extracted using 3 mL of 80% ethanol. The suspensions were vortexed for 15 min and centrifuged at 1500× *g* for 10 min [58]. Supernatants were decanted into 1.5 mL safe-lock microcentrifuge tubes, labelled, and frozen at −20 °C until further analysis.

Immunoreactive fGCM concentrations were quantified using a corticosterone-3-CMO enzyme immunoassay (EIA) [56,59] according to procedures described by Ganswindt et al. [60]. Detailed assay characteristics, including full descriptions of the assay components and antibody cross-reactivities, are provided by Palme and Möstl [59]. The sensitivity of the EIA used at 90% binding was 3.6 ng/g faecal dry weight (DW). Interassay coefficients of variation (CV), determined by repeated measurements of low- and high-quality controls, were 11.74% and 12.91%, respectively, and intraassay CV were 5.59% and 6.61%, respectively. Faecal steroid concentrations are presented as µg/g faecal DW.

### 2.6. Data Preparation

#### 2.6.1. Behavioural Observations

The frequency with which each cheetah in the study performed each behaviour during each observation session was calculated as a proportion of the total number of scan samples carried out during that observation session per study cheetah [48,49]. The resulting data highlighted the proportion of scan samples in which each behaviour was observed during the treatment and control and on feeding and fasting days for each study cheetah.

#### 2.6.2. Faecal Consistency Scores and Glucocorticoid Metabolite Concentrations

Cheetahs typically defecate once daily, attenuating diurnal and pulsatory glucocorticoid (GC) secretion variations in the faeces [61]. However, differences in species and individual traits can affect hormone concentrations and GI transit time [62]. In a study by Terio et al. [42], peak concentrations of GC metabolites were found in the first faecal sample collected from cheetahs after administering adrenocorticotropic hormone. This was comparable to domestic cats’ faecal cortisol excretion rate [63]. This study assumed FCS and fGCM concentrations to reflect the previous day’s intervention to account for cheetahs’ specific 24 h gut passage rate and excretory pattern. The within-subject experimental design, where each cheetah served as their own control, eliminated interindividual variability [64].

### 2.7. Statistical Analysis

Statistical analysis was performed using Microsoft Excel (version 16.0) and JMP Pro software (version 16.0) for Windows, developed by SAS Institute Inc (Cary, NC, USA). The variables measured were explored for univariate outliers greater than three interquartile ranges (IQR) away from the 99.5th or 0.05th percentiles. No outlying values were detected. Normal distribution and homogeneity of variance were explored using Anderson-Darling [65] and Levene’s tests [66], respectively. Behaviour, FCS, and fGCM concentration data were Box–Cox transformed [67] to satisfy the assumption of normality and homogeneity due to their departure. The data were back-transformed for descriptive statistics and visual representation to maintain statistical integrity. In this study, a mixed model for repeated measures (MMRM) analysis [68] was used to investigate the independent fixed effects of the study period and feed versus fast day on (i) behaviour, (ii) FCS, and (iii) fGCM concentration and their interaction to test the moderator effect of the study period on feed versus fast day. An MMRM analysis investigated the independent fixed effect of treatment week (one, two, and three) on (i) behaviour, (ii) FCS, and (iii) fGCM concentration. The study cheetah was included as the random effect in the analyses.

Multiple pairwise comparisons were explored using Tukey’s honestly significant difference (HSD) post-hoc tests [69]. Effect sizes of pairwise comparisons were calculated using the following formula:*d* = *x*_1_ − *x*_2_/√*SD*_1_^2^ + *SD*_2_^2^/2,(1)
where *d* = Cohen’s d effect size; *x*_1_ and *x*_2_ = means of the two groups; and *SD*_1_ and *SD*_2_ = standard deviation of the two groups [70]. Root mean square standardised effects (RMSSE) were interpreted as small (*d* = 0.2), medium (*d* = 0.5), and large (*d* = 0.8) based on Cohen’s d effect size criteria. Descriptive statistics were reported as median (IQR), and the significance level, alpha, was set at 0.05.

## 3. Results

### 3.1. Behavioural Observations

Three hundred and sixty scan samples per study cheetah were collected during the treatment (*n* = 180) and control (*n* = 180). The study cheetahs spent most of their time inactive (for the treatment: 40.37% (week one: 37.02%, week two: 40.61%, and week three: 42.13%) and control: 43.72%, and for feeding: 40.11% and fasting days: 51.30%) (Appendix A). The MMRM analysis revealed that the fixed effects of the study period and feed versus fast day on each behaviour failed to achieve statistical significance; therefore, post-hoc testing was not performed.

The MMRM analysis revealed that the fixed effect of treatment week on locomotion was significant (*F*_3,99.87_ = 2.90, *p* = 0.037). Post-hoc comparisons using Tukey’s HSD test revealed that locomotion was significantly higher (*t*_99.9_ = 2.94, *p* = 0.021) during week three of the treatment (24.84%) than the control (14.80%) (Appendix A).

Effect size calculations using Cohen’s d revealed a medium RMSSE (*d* = 0.77; *t*_101_ = 2.99, *p* = 0.004) on locomotion between week three of the treatment and the control (Table 4).

### 3.2. Faecal Consistency Scores

Two hundred and thirteen faecal samples were collected from the study cheetahs during the treatment (*n* = 105) and control (*n* = 108). The soft with shape faecal grade was the most frequently recorded in the study cheetahs (for the treatment: 45.71% (week one: 45.45%, week two: 45.95%, and week three: 45.71%) and the control: 48.15%, and for feeding: 48.81% and fasting days: 40.00%) (Appendix A). The MMRM analysis revealed that FCS was significantly higher (*F*_1,205.2_ = 10.22, *p =* 0.002) during the treatment (3 (2)) than the control (3 (1)). The MMRM analysis revealed that the fixed effect of feed versus fast day on FCS failed to achieve statistical significance. Post-hoc comparisons using Tukey’s HSD test revealed that FCS was significantly lower on control fasting days (2.5 (1.25)) than on treatment feeding days (3 (1.25); *t*_205.4_ = −2.96, *p =* 0.018), treatment fasting days (4 (2); *t*_205.4_ = −3.31, *p =* 0.006), and control feeding days (3 (2); *t*_205.4_ = −2.83, *p =* 0.027) (Figure 1).

The MMRM analysis revealed that the fixed effect of treatment week on FCS failed to achieve statistical significance; therefore, post-hoc testing was not performed.

Effect size calculations using Cohen’s d revealed a medium RMSSE (*d* = 0.65; *t*_209_ = 3.31, *p* = 0.001) on FCS between the treatment and control (Table 4). The RMSSE of feed versus fast day on FCS by the study period was large between control fasting days and treatment feeding days (*d* = 1.07; *t*_209_ = 3.17, *p* = 0.002), treatment fasting days (*d* = 1.23; *t*_209_ = 3.43, *p* = 0.001), and control feeding days (*d* = 1.01; *t*_209_ = 3.04, *p* = 0.003).

### 3.3. Faecal Glucocorticoid Metabolite Concentrations

The MMRM analysis revealed that the fixed effects of the study period and feed versus fast day on fGCM concentration failed to achieve statistical significance; therefore, post-hoc testing was not performed (Appendix A).

The MMRM analysis revealed that the fixed effect of treatment week on fGCM concentration was significant (*F*_3,166.3_ = 3.14, *p* = 0.027). Post-hoc comparisons using Tukey’s HSD test revealed that fGCM concentration was significantly higher (*t*_166.3_ = 2.85, *p* = 0.025) during week two of the treatment (1.17 (0.59) µg/g DW) than the control (0.90 (0.43) µg/g DW) (Figure 2).

Effect size calculations using Cohen’s d revealed a medium RMSSE (*d* = 0.58; *t*_170_ = 2.76, *p* = 0.006) on fGCM concentration between week two of the treatment and the control (Table 4).

## 4. Discussion

This study aimed to investigate the responses of captive-born (hand-reared) cheetahs to a reduced feeding frequency. The results of the FCS analysis support, to some extent, the researchers’ hypothesis that the more natural feeding pattern would beneficially impact the GI ecosystem, including the microbial fermentation process. Overall, the findings indicate that the change in feeding frequency did not result in a significant behavioural or physiological stress response, contrary to what was predicted.

Animals’ GI tracts harbour essential gut microbes serving various functions [18]. However, disturbance-related deviation in the microbial diversity and abundance pattern beyond a natural range, i.e., gut dysbiosis, can advance pathophysiology and affect host health [71]. Considering the purported link with intestinal microbiota composition [72,73], higher FCS indicates that less frequent feeding could have benefited the studied cheetahs’ GI health. The data present here is consistent with Altman et al.’s [37] work concerning the impact of a random gorge and fast feeding schedule on the digestion of a horsemeat-based diet in captive lions.

Studies have shown that chronic or repeated exposure to stressors can disrupt gut homeostasis [74,75]; therefore, an alternative interpretation of this result may be the stress-reducing effects of less frequent feeding as a potential form of environmental enrichment (EE). In animal husbandry, the principle of EE is widely used to provide species-appropriate challenges to captive animals that lack adequate stimuli. This encourages them to engage actively with their environments, reducing stress and stereotypical behaviour [76,77]. One common type of enrichment is food-based, which also applies to the cheetah [41,48,49,78,79].

Absent hunting opportunities, offering carnivores predictable, fixed, and small daily meals can worsen their tendency to be inactive in captivity. This can result in obesity and affect their wellbeing [22]. In this study, the cheetahs spent most of their time inactive, consistent with previous research on captive cheetahs [79] and other felids [80,81,82,83,84,85,86]. The reduced feeding frequency schedule resulted in numerically lower inactivity and higher locomotion (the latter significantly so during week three of the treatment). Increased activity has similarly been reported in captive lions following the adaption from a conventional zoo feeding programme to a randomised feeding schedule [37].

Stress reduction using EE extends to sympathoadrenal responses. The two branches: the sympathetic nervous system (SNS) and the hypothalamic–pituitary–adrenal (HPA) axis, work together to maintain or re-establish homeostasis by orchestrating behavioural and physiological adaptations to the stressor [87]. The SNS provides the immediate first wave of the stress response, mediating the rapid release of the catecholamine hormones epinephrine (E) and norepinephrine from the adrenal medulla [88]. The second wave is more gradual and involves GCs, the product of the hormonal cascade along the HPA axis [89]. Multiple hypotheses have been proposed to explain the stress-reducing effects of EE, some of which are based on the contention that EE itself acts as a mild stressor [90]. By providing challenges appropriate to an animal’s sensory, physical, and cognitive capacities [91], EE is thought to enable arousal and activation of the physiological stress response without pushing the animal into high, maladaptive stress levels [92,93]. In this manner, EE is adaptive, improving animals’ capacity to cope with stressors; that is, stress resilience [94,95,96]. Numerically and significantly higher fGCM concentration during weeks one and two of the treatment, followed by lower values during week three, suggests the studied cheetahs experienced an acute stress response to the change in feeding frequency to which they adapted. Having previously been provided EE may have increased the captive-born cheetahs in this study’s resilience to enrichment-induced arousal.

More work remains to be carried out before fully understanding the optimal feeding regime(s) to beneficially modulate cheetahs’ intestinal microbiota composition. This study has at least four potential limitations to be considered. Firstly, faecal consistency scoring lacks specificity as a discrete measure of GI health and should not be interpreted as providing empirical evidence of diet suitability for maintaining cheetahs in captivity. Further research is needed to validate FCS against other measurements of microbiome health effects, such as digestibility, pH, the incidence of vomiting or diarrhoea, veterinary diagnosis of GI disease, fermentation byproducts, faecal frequency, dry weight percentage, and gut microbiota and short-chain fatty acids.

It must also be acknowledged that the present study was conducted on hand-reared cheetahs. In mammals, there is evidence that the high microbial diversity of infants’ gut communities may be inherited from their biological mothers [97]. Studies have shown that the rearing method (hand- versus mother-reared) can affect animal gut microbiota composition [98,99]. Therefore, the findings should be generalised to only some captive cheetahs.

Secondly, the duration of each period, i.e., treatment, control, and washout, was selected to accommodate the inverse relationship between the number of T_b_, HR, and LA recordings made using the biologgers and the lifespan of their batteries (refer to Appendix B). The 3-week treatment and control periods may have needed to be longer to produce a definitive response in the studied cheetahs, limiting the conclusions that can be made from the results. For example, in mice, stress-related alterations in the composition and function of faecal microbiota were described after eight days, while they appeared after ten days in rats [100,101]. It would be recommendable for future research to lengthen the current study period and washout to quantify the treatment effects better and prevent possible carryover effects.

A third potential limitation to consider is GI transit time. Measuring fGCM concentrations provides an integrated measure of adrenocortical activity. It reflects the cumulative secretion of plasma GCs over several hours (6–24 h, depending on species-specific defecation rate), attenuating fluctuations due to secretory patterns [102,103,104,105]. However, dietary intake could affect the faecal excretion of steroid hormone metabolites independent of a stress response [106]. Due to accelerated GI transit time, larger quantities of food may decrease the accumulation time of faeces in the intestine and increase metabolite concentration variability [107].

By design, there were fewer fasting days during the routine feeding schedule than during the reduced feeding frequency schedule. The days on which the studied cheetahs’ behaviour was observed were not adapted to maintain fast-day observation sessions equivalent between the treatment and control; therefore, it is possible that fasting days were over- and underrepresented during the treatment and control, respectively.

## 5. Conclusions

Though the validity of FCS as a measure of GI health must be established by further research, these results provided preliminary evidence for a reduced feeding frequency schedule to mediate the unnatural composition of horsemeat-based diets routinely fed to captive cheetahs and as an effective EE strategy. While previous studies have mainly examined the epidemiological relationship between diet composition and GI disease [5,19,20,22,23], the findings presented here indicate that feeding regimes may also play a significant role. This study expands on existing research by Whitehouse-Tedd et al. [5] in developing a global standard by which captive facilities can score their cheetahs’ faecal consistency.

## Figures and Tables

**Figure 1 animals-13-02783-f001:**
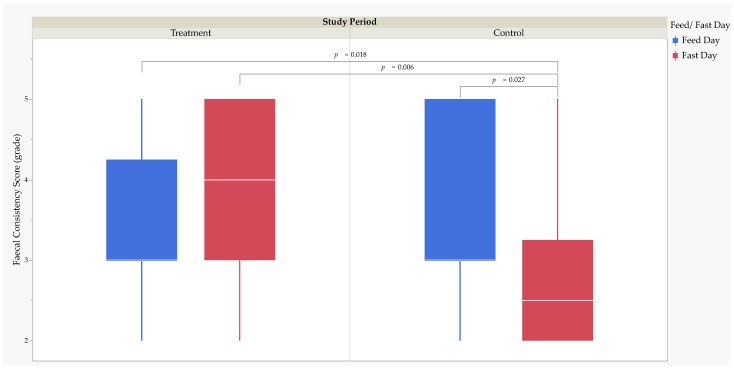
Box and whisker plot of faecal consistency score (grade) for the study cheetahs (CH-2205, -2206, -2207, -2271, -2276, and -2277). Effect of feed versus fast day by the study period. Statistics were performed using Tukey’s honestly significant difference post-hoc test.

**Figure 2 animals-13-02783-f002:**
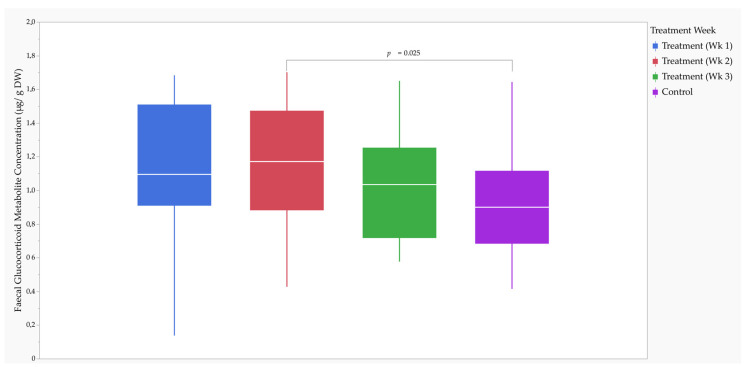
Box and whisker plot of faecal glucocorticoid metabolite concentration (µg/g dry weight (DW)) for the study cheetahs (CH-2205, -2206, -2207, -2271, -2276, and -2277). Effect of treatment week (Wk; one, two, and three). Statistics were performed using Tukey’s honestly significant difference post-hoc test.

**Table 1 animals-13-02783-t001:** Demographic information of six captive-born (hand-reared) cheetahs.

Group	Identification Number	Housing	DOB ^1^ (YY-MM-DD)	BM1 ^2^ (kg)	BM2 ^3^ (kg)	Sex
1	CH-2205	Paired (with CH-2206)	9 June 2016	45.0	45.0	M ^4^
	CH-2206	Paired (with CH-2205)	9 June 2016	47.15	47.65	M ^4^
	CH-2207	Single	9 June 2016	36.6	37.1	F ^5^
2	CH-2271	Single	28 August 2017	37.65	41.75	M ^4^
	CH-2276	Paired (with CH-2277)	16 September 2017	39.9	39.35	F ^5^
	CH-2277	Paired (with CH-2276)	16 September 2017	43.05	43.55	F ^5^

^1^ DOB: date of birth; ^2^ BM1: body mass at the start of the experimental trials; ^3^ BM2: body mass at the end of the experimental trials; ^4^ M: male; ^5^ F: female.

**Table 2 animals-13-02783-t002:** The experimental design used in the study.

Group	Identification Number	Reduced Feeding Frequency Schedule (Treatment)					Washout	Routine Feeding Schedule (Control)				
		Period	Feeding Days	Feeding Time(s)	Amount Fed/Day	Fasting Day(s)		Period	Feeding Days	Feeding Time(s)	Amount Fed/Day	Fasting Day(s)
1	CH-2205	15 May 2019 to 4 June 2019	Mon ^1^, Tues ^2^, Thurs ^3^, Fri ^4^	0800–1700 h	2.7 kg	Wed ^5^, Sat ^6^, Sun ^7^	1 day	6 June 2019 to 26 June 2019	Mon ^1^, Tues ^2^, Wed ^5^, Thurs ^3^, Fri ^4^, Sat ^6^	0800–1200 h, 1500–1700 h	1.8 kg	Sun ^7^
	CH-2206											
	CH-2207	23 April 2019 to 13 May 2019					24 days					
2	CH-2271	31 July 2019 to 20 August 2019	Mon ^1^, Tues ^2^, Thurs ^3^, Fri ^4^	0800–1700 h	2.5 kg	Wed ^5^, Sat ^6^, Sun ^7^	1 day	22 August 2019 to 11 September 2019	Mon ^1^, Tues ^2^, Wed ^5^, Thurs ^3^, Fri ^4^, Sat ^6^	0800–1200 h, 1500–1700 h	1.6 kg	Sun ^7^
	CH-2276	9 July 2019 to 29 July 2019					24 days					
	CH-2277											

^1^ Mon: Monday; ^2^ Tues: Tuesday; ^3^ Thurs: Thursday; ^4^ Fri: Friday; ^5^ Wed: Wednesday; ^6^ Sat: Saturday; ^7^ Sun: Sunday.

**Table 3 animals-13-02783-t003:** The behaviours recorded in the study and their definitions.

Inactive	
Inactive	Laying/asleep, laying/awake, sitting (stationary in a bipedal position)
Active	
Individual behaviour	
Appetitive behaviour	Feeding, food anticipatory activity, stalking
Attention	Staring at one area or paying attention to any visual or auditory stimulus
Autogrooming	Licking or scratching of the own body
Environmental enrichment	Interacting with an enrichment device by biting, dragging, scratching, or carrying it in the mouth
Locomotion	Jumping, running, solitary play, walking
Maintenance	Drinking, defecating/urinating, yawning
Olfactory exploration	Sniffing the air, an object, or the substrate; performing flehmen
Scent marking	Marking substrates or objects in the enclosure by urine-spraying (releasing urine backwards against a vertical surface or object while standing with tail raised vertically), rolling, and rubbing (leaving scents on the substrate or on any object, respectively)
Standing	Stationary in a quadrupedal position
Stereotypical	Pacing (repetitive, apparently functionless locomotory movement along a given route uninterrupted by other behaviours)
Vocalisation	Chirping, growling, purring, stutter-barking, or yowling
Social behaviour	
Affiliative behaviour *	Social play (play-fight, chasing, or playing together with an enrichment item), pawing, or rubbing on a conspecific, social grooming (licking a conspecific or being licked), paying attention to conspecifics by observing them with interest, and interacting with human caretakers
Agnostic behaviour *	Aggression, dominance mount, threat display
Interspecific behaviour	Paying attention to another species’ presence
Not observed	
Out of sight	Focal animal is not visible from the point of observation/behaviour unknown

Informed by the previous literature on felid behaviour (specifically cheetahs) [37,49,50] and initial observations of the cheetahs being studied. * Includes actions performed or received by the focal animal.

**Table 4 animals-13-02783-t004:** Root mean square standardised effects on behaviour, faecal consistency score (grade), and faecal glucocorticoid metabolite concentration (µg/g dry weight (DW)) for the study cheetahs (CH-2205, -2206, -2207, -2271, -2276, and -2277). Effect sizes were calculated using Cohen’s d.

Variable	Effect	Level	-Level	*t*	*p*	df ^1^	Cohen’s d	95% CI ^2^ for Cohen’s d	
								Lower	Upper
Behaviour									
Locomotion	Treatment Wk ^3^	Treatment (Wk ^3^ 3)	Control	2.99	0.004	101	*0.77*	0.25	1.29
Faecal consistency score (grade)	Study period	Treatment	Control	3.31	0.001	209	*0.65*	0.26	1.03
	Study period*feed/fast day	Treatment, fast day	Control, fast day	3.43	0.001	209	**1.23**	0.51	1.94
	Study period*feed/fast day	Treatment, feed day	Control, fast day	3.17	0.002	209	**1.07**	0.40	1.74
	Study period*feed/fast day	Control, feed day	Control, fast day	3.04	0.003	209	**1.01**	0.35	1.66
Faecal glucocorticoid metabolite concentration (µg/g DW)	Treatment Wk ^3^	Treatment (Wk ^3^ 2)	Control	2.76	0.006	170	*0.58*	0.16	1.00

Numbers in italics represent a medium magnitude of effect (*d* = 0.5), while bold numbers represent a large magnitude of effect (*d* = 0.8). ^1^ df: degrees of freedom; ^2^ CI: confidence interval; ^3^ Wk: week.

## Data Availability

The data presented in this study are available in Appendix A.

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
