# Peer review of "Responses to Reduced Feeding Frequency in Captive-Born Cheetahs (Acinonyx jubatus): Implications for Behavioural and Physiological Stress and Gastrointestinal Health"

_animals, 2023, doi:10.3390/ani13172783_

Round 1
Reviewer 1 Report
Thank you for the opportunity to review this manuscript. The rationale for the study, the methods, and the results are unusually well reported—kudos to the authors. Provided below are a few minor suggestions for improvement when it comes to things that can be changed. My main concerns center around things that cannot be changed.
First is the study design, and the possible effect of treatment order on the results. The duration of each study phase is unclear, and this information is important to interpreting the results. The simple summary (but not the methods) says “three weeks,” but as phrased I’m unsure whether this means a total of three weeks or three weeks per study phase. The discussion (line 579) says “3-week treatment and control periods,” so on reviewing this, I’ve assumed each phase was three weeks long. This information needs to be better spelled out in the methods and elsewhere. Also, I would think that the stress of a feeding change might influence the results, particularly for the first few days of each phase. The authors—as reported--looked at differences within days and between treatment and starve/feed days, but not at differences over time within each phase—which might be important. Further, I have concerns that only one day separated the treatment and control phases, and that all 6 cheetahs underwent the same phase order. This could have allowed possible carryover effects from the treatment phase, confounding findings for the second phase. Ideally, the effect of order would have been better controlled for (e.g., by starting half of the cheetahs with the control protocol).
Another concern is the small number of animals with useful biologger data, which might have yielded significant differences despite the sample size (likely due to the large number of measurements, with more than 30,000 degrees of freedom—Figures 4 and 5 nicely demonstrate this concern) but may not be clinically meaningful or widely generalizable. The authors are transparent about this in the discussion of limitations, but this doesn’t fix the problem.
Finally, I’m generally of the mindset that any data regarding the issues captive cheetahs struggle with are important to share, despite any study flaws, provided care is taken to not overstate the conclusions. To this end, I believe the conclusions (lines 613 to 625) are unsupported by the findings/study design. Did the authors really demonstrate the usefulness of biologger technology? I‘m not convinced and, in fact, concluded the opposite based on what was reported. Did they find “practical and theoretical welfare implications for cheetahs in captivity?” I’m not convinced of that either. And I disagree that the data provided “sufficient support for a reduced feeding frequency schedule” with all the limitations in mind. Perhaps they did offer preliminary evidence of this.
Suggested changes
Overall: This is a lengthy paper, dense with information that may or may not be essential to include in print (vs as supplementary material). Considerable space is given to biologger results, when the authors acknowledge in the discussion that these data “were likely irrelevant to the biology of the cheetahs studied.” Much less space is given to the results that matter more (e.g., fecal consistency). Please reconsider whether it’s helpful to repeat results in both the narrative and in figures or tables, and please review the figures that have been included for helpfulness to readers for interpreting the results.
Consider changing the title of Table 1 to incorporate the data pertaining to all cheetahs [captive-born (hand-reared)] and removing the column containing that information. For example, a new title might be “Demographic information of six captive-born (hand-reared) cheetahs.”
The intended meaning of “(±500 m)” on line 142 is unclear. It’s the ± part that’s unclear. Was this intended to instead say “within 500 m”? Kindly rephrase for clarity.
Experimental design section: Please consider spelling out or making more prominent the precise duration of each phase of the feeding trial. I struggled to find this information, and it’s important for assessing study validity.
End parenthesis missing at line 211.
At line 232 (title of Table 2), I wonder if the word “ethogram” is what’s meant, as I believe an ethogram refers to the actual descriptions/observations of subjects, and not to the identities of measured elements. Perhaps the title could be rephased to remove “Ethogram of.”
Line 249: Is this true: “(grade 4: firm and dry and grade 5: firm)”? Could it be that grade 5 was actually “extremely dry”?
Statistical analysis: Generally strong, although again I wonder about whether it would’ve been more prudent to control for study day rather than starve/feed day.
Line 380: Two observations:
1) The word “the” is throwing me in this footnote. This might be easier to interpret if rephrase “Numbers in italics represent a medium magnitude of effect, while bold numbers represent a large magnitude of effect.”
2) Maybe it’s just my eyes, but I don’t see any bolded Cohen’s d values.
Results
Consider rounding all P values to no more than 3 decimal places, as additional decimal places add no useful information and give the illusion of things being more significant than they were.
Reviewer 2 Report
I will upload the document with my comments. I recommend that throughout the entire document you change "starve" to "fast" or "fasting" as the word starve connotates a very negative action and poor animal welfare considerations. Whereas zoos and sanctuaries use "fasting" days routinely for big cats and other carnivore, they don't say "starve" which the public misconstrues for terrible animal care. There are also some questions where an "n" of 3 can really lead you statistically to some of your conclusions.
Assessing gut microbiome was not done. To do this, you really should do next generation PCR which from a fecal swab determines all bacteria, yeast, fungi within it. This could have been done concurrently to see if the changes in feeding makes any difference. Take a look at the company MiDog and contact Dr. Janina Krumbach for further information. www.midog.com I do not work for them, but have been using their services now to look at non-dog/non-cat species medical issues.
Fecal steroid conclusions - again, you need to look at changes throughout the year to assess much. As you've also stated - there were no measurable changes. You are working with animals accustomed to having humans around, changes in their environment, routines etc so it could be very different with wild cheetahs.
I have made notations within the pdf. 2 references need to have the titles fixed.

just changing "starve" to "fast" or "fasting" which from an animal welfare issue is far more acceptable and commonly used in working with big cats
Round 2
Reviewer 1 Report
The authors have addressed my concerns to my satisfaction. Thank you for your hard work!